# Evaluating Fatalism Among Breast Cancer Survivors in a Heterogeneous Hispanic Population: A Cross-Sectional Study

**DOI:** 10.3390/curroncol32080461

**Published:** 2025-08-15

**Authors:** Liara Lopez Torralba, Brian Sukhu, Maria Eduarda de Azevedo Daruge, Jongik Chung, Victoria Loerzel, Eunkyung Lee

**Affiliations:** 1Burnett School of Biomedical Sciences, College of Medicine, University of Central Florida, Orlando, FL 32816, USA; li650169@ucf.edu; 2Department of Health Sciences, College of Health Professions and Sciences, University of Central Florida, Orlando, FL 32816, USA; brian.sukhu@ucf.edu (B.S.); maria.eduardadeazevedodaruge2@ucf.edu (M.E.d.A.D.); 3Department of Statistics and Data Science, College of Sciences, University of Central Florida, Orlando, FL 32816, USA; jongik.chung@ucf.edu; 4Department of Nursing Practice, College of Nursing, University of Central Florida, Orlando, FL 32827, USA; victoria.loerzel@ucf.edu

**Keywords:** breast cancer, fatalism, Hispanic, cancer survivorship, disparity, country of origin

## Abstract

A key factor affecting the quality of life among Hispanic breast cancer survivors is fatalism, the belief that events are predetermined and beyond personal control. Our study in Central Florida found that these survivors exhibited moderate fatalism, often linked to strong religious beliefs and an internal locus of control. Individuals from Dominican, Mexican, and Venezuelan backgrounds showed higher fatalism, while those with higher incomes, more education, and greater English use reported lower levels. Healthcare providers should recognize patients with higher fatalism and adapt their approaches accordingly. These insights can help develop culturally sensitive interventions and support strategies to enhance patients’ sense of control

## 1. Introduction

Breast cancer is the leading cancer diagnosed among Hispanic/Latina women in the United States (US) [1]. Although they are less likely to be diagnosed with breast cancer and die from this disease compared to their non-Hispanic White counterparts [2], they face unique challenges. These include diagnoses of high-grade tumors at younger ages [3], which entails a worse prognosis. Latina breast cancer survivors report experiencing more severe symptoms, such as fatigue, depression, and difficulties in work and spousal relationships [4,5,6] and a lower overall quality of life (QOL) compared to non-Hispanic survivors [7,8,9,10].

Growing evidence indicates that cultural factors play a significant role in cancer survivorship, particularly concerning QOL disparity [9,11]. Fatalism has been recognized as a dominant belief among Hispanic populations compared to non-Hispanic Whites [12]. Research suggests a substantial association between fatalistic beliefs and adverse outcomes in overall well-being and concerns related to breast cancer survivorship [13,14]. Fatalism, as defined by Merriam-Webster, is the belief that events are predetermined, leaving individuals powerless to change them [15]. This viewpoint implies that people have limited control over their fate concerning cancer incidence or survival [16]. High levels of fatalism reflect a diminished sense of personal control over health outcomes, whereas low levels indicate a greater sense of autonomy [17]. Research shows that fatalism can impede proactive behaviors, such as breast cancer screening and timely medical responses [18,19].

Examining fatalism is crucial, particularly as the Hispanic population continues to grow. Some studies showed that several factors, including acculturation, age, education, religion, locus of control, poverty, and cancer fear, influence fatalism among cancer survivors [20,21]. However, the findings are inconsistent across different study populations. For example, a study of ethnic minorities in the United Kingdom found a negative association between fatalism and fear of cancer [20], while another study involving Chinese breast cancer survivors found a positive association with fear of recurrence [22]. Research on fatalism among Latina breast cancer survivors is limited, as previous studies either excluded Hispanic populations from the study sample or did not explicitly focus on fatalism as the key study variable [16,21,23,24,25]. Additionally, studies did not adequately differentiate between various Hispanic subgroups [7,16,21].

Thus, the main objectives of the present study were twofold. First, it aimed to explore the factors independently associated with fatalism among Hispanic/Latina breast cancer survivors to identify individuals who are at higher risk for experiencing high fatalism. Secondly, the study examined how fatalism varies according to different Hispanic backgrounds. By understanding these perspectives, healthcare providers can develop tailored, culturally responsive strategies to reduce fatalism in these populations.

## 2. Materials and Methods

### 2.1. Study Design and Population

This is a report of an ongoing population-based, cross-sectional study that aims to evaluate the mediators of QOL disparities among breast cancer survivors within the Hispanic community in Central Florida, US. The Florida Cancer Data System (FCDS) provided contact information for individuals who met the eligibility criteria of the study. The eligibility criteria required participants to be adult women presumed to be Hispanic or Latina, aged ≥ 20 years, diagnosed with breast cancer from 2015 to 2023 in Central Florida, and reported to the FCDS. We followed the state-mandated patient recruitment procedure [26]. Our success in this recruitment process has been reported elsewhere (in press) [27]. Briefly, bilingual invitation letters were sent to all individuals deemed eligible to gauge their interest in participating in the study. If there was no response after three weeks, a second letter, accompanied by a telephone opt-out card, was sent to facilitate their participation. If there was no response to the second invitation, the bilingual research team made follow-up phone calls, attempting up to four times to gauge prospective participants’ interest in the study.

Among the 3398 women to whom we sent invitation letters, a total of 1234 women were found to be contactable and eligible for the study. We sent a survey packet to 720 women who expressed interest in participating, either online or via mail, depending on their choice, in their preferred language of English or Spanish. The survey comprised questionnaires measuring QOL, unmet needs, sociodemographic characteristics, lifestyle, comorbidities, social support, fatalism, life satisfaction, and breast cancer worry. Between September 2023 and January 2025, a total of 413 women returned the survey. We excluded two participants who identified as non-Hispanic and 21 individuals due to missing fatalism scores. This results in a final sample of 390 women for data analysis in the current study. All participants provided informed consent for their involvement and received a USD 25 gift card as compensation for their participation. The study was approved by the University of Central Florida Institutional Review Board (IRB) and the Florida Department of Health IRB. The detailed numbers are displayed in Figure 1.

### 2.2. Measures

#### 2.2.1. Fatalism

The Multidimensional Fatalism Measure (MFM) was used to assess fatalistic beliefs. This questionnaire was developed and validated in both English and Spanish, ensuring linguistic consistency across different populations, and it showed strong psychometric properties, including high reliability and validity [28]. The MFM consists of 30 questions, assessing general fatalism (e.g., if something bad is going to happen to me, it will happen no matter what I do) and four related domains, including divine control (e.g., whatever happens to me in my life, it is because that is the way God wanted it to happen), internality (e.g., what happens to me is a consequence of what I do), helplessness (e.g., I feel that nothing I can do will change things), and luck (e.g., some people are simply born lucky). Each domain has six items on a five-point Likert scale ranging from 1 (not at all) to 5 (very much), with the highest score possible in each domain being 30. A high fatalism score indicates a high belief in general fatalism. A high score in divine control indicates high religious beliefs, while a high score in luck indicates a high belief in luck. A high score in internality indicates a strong internal locus of control, and a high score in helplessness indicates a strong external locus of control. Since there are no cut-off points suggested for the level of belief, we used scores of 6–12, 12–21, and 21–30 to define the levels of belief as low, moderate, and high, respectively, to enhance understanding of the characteristics of fatalism in the current study. Although there are several instruments to measure fatalism, including the Powe Fatalism Inventory or the Breast Cancer Fatalism Scale, we chose the MFM instrument, because MFM can assess multiple dimensions of fatalism, which allows for a more nuanced understanding of fatalistic beliefs [28] compared to other instruments, which are unidimensional, measuring only fatalistic belief for a specific disease outcome [29].

#### 2.2.2. Hispanic Origin

Hispanic origin was self-reported in the survey, and this information was also collected by the cancer registry. Whenever possible, participants were categorized according to their self-reported origin. For the 19 women who did not provide a self-report of their Hispanic origin, supplementary information regarding nationality and Hispanic origin in the FCDS dataset was used. Hispanic origin was classified by country, which included Colombian, Cuban, Dominican, Mexican, Puerto Rican, Venezuelan, and Other Hispanic. Origins with <20 participants were grouped together under the ‘Other Hispanic’ category due to the low sample size.

#### 2.2.3. Acculturation

As acculturation is seen as adapting to the culture and norms of a new environment surrounding a person [30], in our case, it would be Hispanic/Latino participants adopting more American views and beliefs. As language proficiency and time spent in a new culture are common predictors of acculturation [31], two questions from the Behavioral Risk Factor Surveillance System [32] were used to measure acculturation: (1) ‘which language do you use at home (only Spanish, more Spanish than English, equal Spanish and English, more English than Spanish, and only English spoken at home)?’ and (2) ‘how long have you lived in the US (in years)?’ Although some studies used less than 10 years, 10–19 years, and 20 or more years as the cut-off [16,33], we grouped the duration of residence into less than 10 years, 10–29 years, and 30 or more years because more than 50% have lived in the US for more than 30 years in the present study.

#### 2.2.4. Fear of Recurrence

The Lerman Breast Cancer Worry Scale is a three-item questionnaire used to measure the level of breast cancer recurrence worry and its impact on daily lives [34]. For the current study, the question, ‘how worried are you about getting your breast cancer back?’ was used to assess the participants’ level of fear of breast cancer recurrence, with categories of 1 (low), 2–3 (moderate), and 4–5 (high).

#### 2.2.5. Other Covariates

In addition to the above characteristics, age at the time of survey completion, race, smoking, marital status, household income, education level, height, weight, chronic medical conditions, and place of birth were self-reported in the survey. Self-reported comorbid conditions included hypertension, cardiovascular diseases, stroke, gastrointestinal problems, chronic lung diseases, arthritis, sciatica, diabetes, osteoporosis, and depression, and the presence of multimorbidity was defined as having two or more comorbidities [35], excluding cancer. The body mass index (BMI) was calculated with reported height and weight. Cancer-related characteristics, including age at diagnosis, diagnosis date, cancer stage, treatments (surgery, radiation, chemotherapy), and status of estrogen receptor (ER), progesterone receptor (PR), and human epidermal growth factor receptor 2 (HER2) were used from the FCDS data.

### 2.3. Statistical Analysis

Descriptive statistics (mean ± SD for continuous variables; frequencies and percentages for categorical variables) were used to describe characteristics of study participants according to their Hispanic origin. The association between fatalism scores and each of the study characteristics was examined with the least squares mean and 95% confidence interval (CIs) using a generalized linear model. A correlation matrix was conducted to examine the interrelationship among different characteristics of fatalism (i.e., general fatalism, religious belief, belief in luck, and internal and external locus of control). To describe the characteristics of fatalism according to Hispanic origin, a radar plot in Microsoft Excel was used to visualize the distribution of the five domains. A multivariable linear regression model was built with variables found significant in the univariate analyses alongside the potential confounders known from the literature review. The normality assumption of fatalism scores was tested with graphical examinations. As shown in Appendix A, the residuals in the multivariable linear model appear to be symmetrically distributed with a bell-shaped pattern, suggesting approximate normality. For missing information found in 11 participants (i.e., education [*n* = 1], language use at home [*n* = 2], fear of recurrence [*n* = 7], and years lived in the US [*n* = 1]), we employed the discriminant function methods for multiple imputations [36] since all of the missing variables were categorical. We performed 250 burn-in iterations to generate 1000 imputed datasets and subsequently estimated regression coefficients for each one. The combined results from these imputed datasets were summarized and reported as coefficients, standard errors, and associated *p*-values. Additionally, a sensitivity analysis was conducted using cases with complete data (list-wise deletion) to assess the robustness of the research findings. Statistical analyses were performed using SAS (version 9.4), and a *p*-value < 0.05 was considered statistically significant.

## 3. Results

### 3.1. Study Participants

The study consisted of 390 participants, and they were grouped based on their self-reported Hispanic origin: Colombian (8.7%), Cuban (6.4%), Dominican (7.4%), Mexican (5.6%), Puerto Rican (53.8%), Venezuelan (6.2%), and Other Hispanics (11.8%), including Peru, Brazil, Honduras, and other countries. Sociodemographic characteristics by Hispanic origin group can be found in Table 1. The mean age of the participants was 59.5 (±11.8) years, with an average of 4.8 (±2.0) years since diagnosis. About half of the participants self-reported their race as White (46.9%) and had a household income of USD 20,000–USD 75,000 (45.1%), while 60% were married and 68.5% had an education of some college or higher. Only 25.4% used more English than Spanish at home, and 79.0% were born outside of the US and have lived in the US for an average of 32.9 (±19.4) years. Most of them were diagnosed at an early stage with ER-positive, PR-positive, or HER2-negative tumors and received surgery, while only less than half of them received radiation (43.6%) and chemotherapy (44.1%) for breast cancer. About 65% showed a moderate or high level of fear of recurrence. Interestingly, Cubans have lived in the US for a longer time than other subgroups (45.4 years), and 48% use English more often at home. A higher proportion of Cubans had an income level of USD 75,000 or more (44.0%), but a smaller proportion had received chemotherapy (28.0%) than other subgroups. In contrast, Venezuelans have lived in the US for the shortest time of 13.5 years. Notably, 83.3% Venezuelans had higher levels of education, 91.7% use Spanish more often at home, and 75.0% have received chemotherapy.

### 3.2. Fatalism Scores by Participant Characteristics

The estimated mean fatalism score was 16.4 (95% CI = 15.8–17.0), as shown in Table 2. The mean scores varied by Hispanic origin, approaching statistical significance (*p* = 0.060). Notably, higher fatalism scores were reported among Dominican, Mexican, and Venezuelan groups, with scores of 17.4, 17.8, and 17.4, respectively, whereas Colombians reported the lowest score of 13.3. Participants with higher income levels (*p* = 0.004), higher education attainment (*p* = 0.003), an increased use of English at home (*p* = 0.007), recent diagnoses within two years (*p* = 0.043), or lower levels of fear of cancer recurrence (*p* = 0.039) reported lower fatalism scores. Fatalism scores did not differ significantly by age groups, race, BMI, smoking, marital status, place of birth, the presence of multimorbidity, cancer stage, and treatments received, or duration of residence in the US.

### 3.3. Characteristics of Fatalism

As shown in Figure 2 and Table 3, Hispanic breast cancer survivors endorsed a strong internal locus of control (23.4 ± 5.0) and religious beliefs (21.3 ± 7.5). They also demonstrated a moderate belief in luck (12.6 ± 4.7) and a low external locus of control (10.9 ± 5.3). The fatalism score was positively correlated with all other measures. As anticipated, fatalism had a stronger correlation with external locus of control and religious beliefs compared to internal locus of control and belief in luck. However, no significant correlations were found between internal locus of control and any measures, except for fatalism.

Figure 2 presents a radar plot illustrating the dynamics of MFM based on Hispanic origin. The mean fatalism scores for Dominicans, Mexicans, and Venezuelans were similar; however, their fatalism characteristics differ. For example, all three groups reported a strong internal locus of control. Dominicans reported high religious beliefs but moderate beliefs in luck and external locus of control. Mexicans reported moderate religious beliefs, beliefs in luck, and an external locus of control. Venezuelans reported strong religious beliefs, moderate beliefs in luck, and low external locus of control. In addition, Colombians and Cubans had a strong internal locus of control, moderate religious beliefs, and low beliefs in luck and external locus of control.

### 3.4. Factors Associated with Fatalism

The multivariable linear regression model included Hispanic origin, current age, race, income level, education level, language use at home, years lived in the US, years since diagnosis, and the level of fear of recurrence, as outlined in Table 4. This model accounted for 14.0% of the variance in fatalism (R^2^ = 0.140, F (22, 367) = 2.7, *p* < 0.0001). Relative to the Puerto Rican subgroup, Colombians reported significantly lower fatalism score, with a mean difference of 3.7 points (β = −3.7, *p* = 0.001). No other subgroups demonstrated significant differences in fatalism scores when compared to Puerto Ricans. Furthermore, participants making between USD 20,000 and USD 74,999 and over USD 75,000 had significantly lower fatalism scores (β = −2.4, *p* = 0.007 and β = −254, *p* = 0.029, respectively) compared to participants with incomes below USD 20,000. Individuals possessing some college education or a higher degree had significantly lower fatalism scores (β = −2.0, *p* = 0.008) compared to those with an education level of high school or below. Participants who primarily use English at home reported a significantly lower fatalism score (β = −1.9, *p* = 0.043) than those who predominantly use Spanish. There were no significant associations between fatalism scores and current age, race, years lived in the US, and the level of fear of recurrence after adjusting for confounding effects. The results from the sensitivity analysis presented in Appendix A align closely with those obtained from the primary analysis shown in Table 4, indicating that the results remain robust.

## 4. Discussion

This study evaluated fatalism among 390 breast cancer survivors of various Hispanic backgrounds. The results demonstrated that these women had moderate levels of fatalism, characterized by a strong internal locus of control and religious belief, a moderate belief in luck, and a low external locus of control. Additionally, breast cancer survivors of Colombian origin and those with higher income, higher education, or proficiency in English reported lower fatalism, highlighting the need for culturally tailored interventions aimed at enhancing cancer survivorship among Hispanic breast cancer survivors.

Hispanic origin significantly influences levels of fatalism, as shown in a previous study where Mexicans and Puerto Ricans had a more fatalistic (negative) view of cancer [37]. In our study, Colombians reported a lower fatalism score compared to Puerto Ricans, contrasting with Diaz’s research, which suggested Colombians have less power to control their health stemming from a collectivist perspective [38]. The Colombians in our study were immigrants or descendants of Colombian Americans who had adapted to US culture, likely leading to a greater sense of control over their health and lower fatalism, as well as a strong internal locus of control. Although several studies reported differences in fatalism between Caucasians and African Americans [12,39,40], we were not able to observe this difference, which could be because many participants in the current study responded either ‘other’ or ‘prefer not to report,’ with only a few identifying as Black, Asian, Native Indian, or Pacific Islander. According to the Pew Research Center, only about 44% of Hispanics see their identity reflected very well with the traditional categories of the race and ethnicity questions in the US Census, and see origins as central to their identity [41]. These findings underscore the importance of considering patients’ origins, including their ethnic backgrounds, nationality, and culture, to enhance understanding of their health beliefs and behaviors, highlighting the need for cultural competency in healthcare.

Income and education have demonstrated a negative association with fatalism scores, as shown in prior research [23,39,42]. This association could be because individuals with higher household incomes often feel a greater sense of control over their medical care and have access to a more extensive support network compared to those with lower incomes [43]. Additionally, individuals with higher levels of education typically have an enhanced understanding of the available treatment options, their associated outcomes, and effective coping mechanisms [44].

The level of acculturation may be linked to fatalism [42,45]. We assessed acculturation by employing two proxy questions: proficiency in language use and duration of residence. Of these variables, only English language proficiency showed a negative association with fatalism, aligning with previous research findings [16,20]. One potential explanation for the lack of association with the length of residence in the US is that most participants in our study have resided in the country for an extended duration, resulting in minimal variability. Acculturation is defined as the process of adapting to the cultural norms and values of a new environment [30]. Therefore, measuring acculturation solely through language proficiency and duration of residence may not provide a complete representation of acculturation. Future studies should consider participants’ cultural norms, values, and practices to offer a more nuanced understanding of acculturation [46], especially when the majority of the study population consists of long-term immigrants.

Although it did not reach statistical significance, there was a slight increase in the mean fatalism score with increasing levels of fear of cancer recurrence, as demonstrated in previous research [22]. Fear of recurrence is common among breast cancer survivors, particularly those with fatalistic beliefs, and it mediates the association between fatalism and anxiety and depression.

It is noteworthy that the Hispanic breast cancer survivors in our study displayed a strong internal locus of control alongside a strong religious belief. This combination suggests that their sense of fatalism may not be solely linked to a belief in a divine being. While Hispanic survivors acknowledge that God controls every aspect of life, they also display an increased sense of personal control in managing their own lives. Stronger fatalistic beliefs, especially when coupled with medical mistrust, can lead to avoidance of necessary medical procedures [21]. Therefore, it is crucial to approach patients who display high fatalism with an understanding of the origins of these beliefs to enhance adherence to screening and treatment [21,39]. By fostering trust between patients and physicians, patients are more likely to discuss their concerns or symptoms earlier in their breast cancer survivorship care. As a result, they may be more inclined to follow recommended treatment plans, ultimately improving their prognosis and QOL. Furthermore, for those with high fatalism, interventions focusing on enhancing perceived control and reducing reliance on fatalistic beliefs could also be beneficial [22].

This study has some limitations that warrant consideration. It was conducted within a specific geographic area, which may accurately reflect the experiences of all Hispanic breast cancer survivors. However, the distribution of Hispanic origins in the sample aligns closely with the demographics observed in Central Florida, as previously reported (in press), thereby rendering the results applicable to the Hispanic breast cancer population in that region. Additionally, due to the cross-sectional design of the study, a causal relationship among the factors examined cannot be established. There is also a possibility of residual confounding, as not all the variables known to influence fatalism, such as coping mechanisms, religious affiliation, or medical access [19,42,47], were included in the multivariable model.

This study has several strengths. It is one of the first to demonstrate a disparity in fatalism based on country of origin, which is not possible when using an aggregated Hispanic group. Fatalism was measured using a valid instrument (i.e., MFM), ensuring linguistic consistency across different populations [28]. Importantly, this instrument incorporates a multidimensional belief system regarding fatalism, allowing for an evaluation of the interrelationship among various beliefs related to fatalism. The findings demonstrated how these beliefs vary by Hispanic origin, enhancing our understanding of the characteristics of fatalism among Hispanic breast cancer survivors.

In conclusion, Hispanic breast cancer survivors reported a moderate level of fatalism, with significant variations according to their specific Hispanic origin. Notably, Colombians had lower fatalism scores in comparison to Puerto Ricans. In addition, Hispanic breast cancer survivors with higher household income, higher levels of educational attainment, and those who predominantly use English rather than Spanish at home reported lower fatalism. These findings provide valuable insights for healthcare providers in seeking to identify individuals who may be at a greater risk of experiencing high fatalism. By recognizing these factors, healthcare providers can adapt their strategies to mitigate fatalism and enhance patients’ perceived control over their health outcomes and behaviors. We will investigate the role of fatalism as a potential mediator in the disparities of QOL within this Hispanic population in the subsequent phase of our study.

## Figures and Tables

**Figure 1 curroncol-32-00461-f001:**
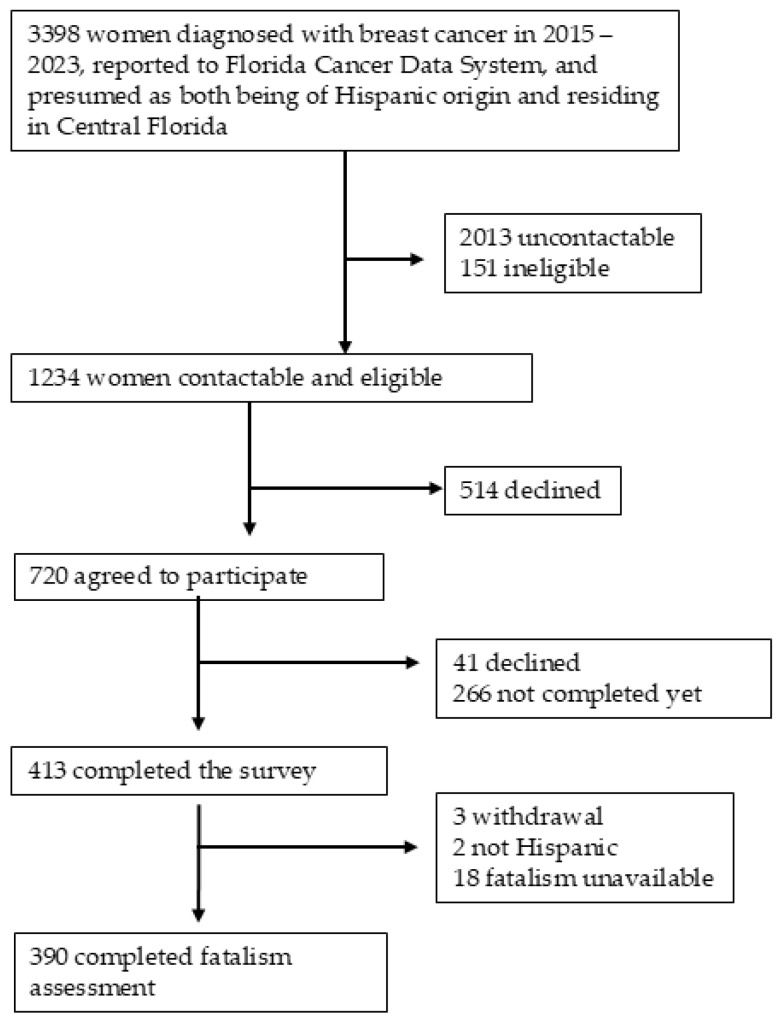
Sample selection process.

**Figure 2 curroncol-32-00461-f002:**
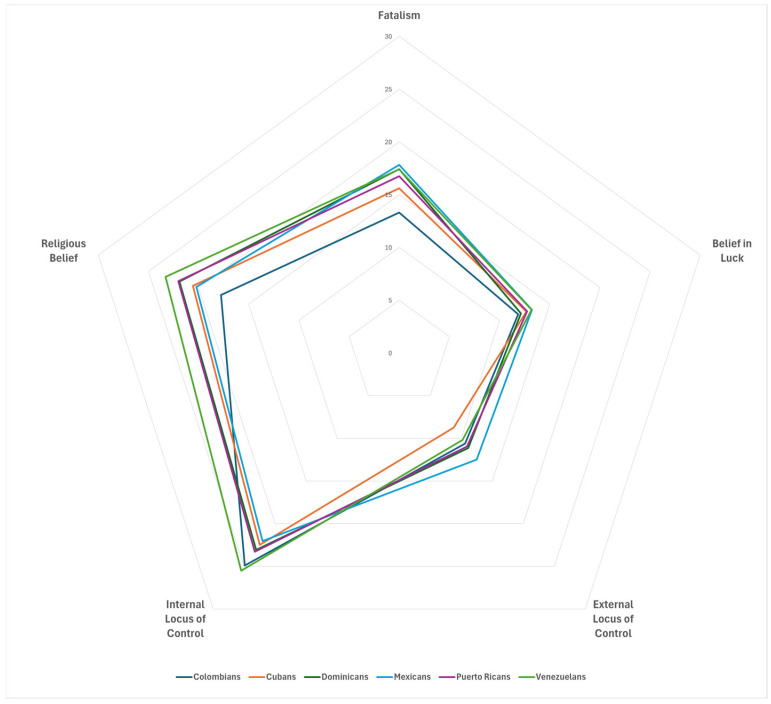
Dynamics of Multidimensional Fatalism Measure of Hispanic breast cancer survivors by Hispanic origin. The radar plot displays the dynamics of multidimensional fatalism measures, starting at 0 in the center and reaching a maximum possible score of 30 at the outer edges. This plot helps us understand the characteristics of fatalism across different Hispanic subgroups and identify subgroups with a lower dimension score.

**Table 1 curroncol-32-00461-t001:** Sociodemographic characteristics of study participants according to Hispanic origin (*N* = 390).

Characteristics/Categories	Total	Colombian	Cuban	Dominican	Mexican	Puerto Rican	Venezuelan	Other Hispanic	
No. of participants	390 (100.0%)	34 (8.7%)	25 (6.4%)	29 (7.4%)	22 (5.6%)	210 (53.8%)	24 (6.2%)	46 (11.8%)	
	Mean (SD)								*p*-value
Age at diagnosis	55.0 (11.9)	56.4 (9.8)	57.1 (12.0)	55.7 (13.3)	50.0 (12.2)	55.7 (11.9)	53.5 (12.3)	52.3 (11.1)	0.177
Current age	59.5 (11.8)	60.4 (9.6)	61.6 (12.3)	60.0 (13.0)	55.0 (11.7)	60.2 (11.8)	57.9 (12.2)	57.6 (11.6)	0.375
Years since diagnosis	4.8 (2.0)	4.8 (2.1)	4.6 (1.9)	4.4 (1.9)	5.1 (2.1)	4.6 (1.9)	4.6 (1.8)	5.6 (2.9)	0.089
Years lived in US	32.9 (19.4)	33.6 (18.2)	45.4 (17.9)	40.9 (16.7)	37.7 (16.5)	31.3 (19.9)	13.5 (11.1)	35.6 (16.1)	<0.001
Body mass index	28.7 (5.7)	26.7 (4.8)	29.3 (5.6)	28.1 (5.4)	29.5 (6.5)	29.2 (5.9)	27.8 (6.2)	27.9 (4.4)	0.194
	*N* (%)								*p*-value
Race									0.029
White	183 (46.9)	15 (44.1)	5 (20.0)	20 (69.0)	12 (54.5)	98 (46.7)	9 (37.5)	24 (52.2)	
Nonwhite	207 (53.1)	19 (55.9)	20 (80.0)	9 (31.0)	10 (45.5)	112 (53.3)	15 (62.5)	22 (47.8)	
Current smoking									0.729
No	372 (95.4)	32 (94.1)	23 (92.0)	29 (100.0)	22 (100.0)	200 (95.2)	23 (95.8)	43 (93.5)	
Yes	15 (3.8)	2 (5.9)	2 (8.0)	0 (0.0)	0 (0.0)	8 (3.8)	1 (4.2)	2 (4.3)	
Marital status									0.074
Married	234 (60.0)	20 (58.8)	20 (80.0)	14 (48.3)	15 (68.2)	117 (55.7)	16 (66.7)	32 (69.6)	
Divorced/separated/widowed	151 (38.7)	14 (41.2)	4 (16.0)	15 (51.7)	6 (27.3)	90 (42.9)	8 (33.3)	14 (30.4)	
Household income									0.090
<USD 20,000	67 (17.2)	10 (29.4)	3 (12.0)	6 (20.7)	2 (9.1)	37 (17.6)	3 (12.5)	6 (13.0)	
USD 20,000–<USD 75,000	176 (45.1)	14 (41.2)	5 (20.0)	11 (37.9)	13 (59.1)	100 (47.6)	13 (54.2)	20 (43.5)	
≥USD 75,000	92 (23.6)	7 (20.6)	11 (44.0)	6 (20.7)	3 (13.6)	48 (22.9)	3 (12.5)	14 (30.4)	
Prefer not to answer	55 (14.1)	3 (8.8)	6 (24.0)	6 (20.7)	4 (18.2)	25 (11.9)	5 (20.8)	6 (13.0)	
Education level									0.025
≤High school	122 (31.3)	13 (38.2)	9 (36.0)	11 (37.9)	13 (59.1)	60 (28.6)	4 (16.7)	12 (26.1)	
Some college+	267 (68.5)	21 (61.8)	16 (64.0)	18 (62.1)	8 (36.4)	150 (71.4)	20 (83.3)	34 (73.9)	
Multimorbidity									0.309
No	157 (40.3)	16 (47.1)	10 (40.0)	11 (37.9)	12 (54.5)	75 (35.7)	12 (50.0)	21 (45.7)	
Yes	218 (55.9)	18 (52.9)	14 (56.0)	17 (58.6)	9 (40.9)	129 (61.4)	11 (45.8)	20 (43.5)	
Language used at home									<0.001
More Spanish	203 (52.1)	22 (64.7)	9 (36.00	13 (44.8)	10 (45.5)	112 (53.3)	22 (91.7)	15 (32.6)	
Both equally	86 (22.1)	5 (14.7)	4 (16.0)	7 (24.1)	1 (4.5)	57 (27.1)	1 (4.2)	11 (23.9)	
More English	99 (25.4)	7 (20.6)	12 (48.0)	9 (31.0)	10 (45.5)	41 (19.5)	1 (4.2)	19 (41.3)	
Birthplace									0.167
US	308 (79.0)	31 (91.2)	18 (72.0)	23 (79.3)	18 (81.8)	159 (75.7)	23 (95.8)	36 (78.3)	
Non-US	81 (20.8)	3 (8.8)	7 (28.0)	6 (20.7)	4 (18.2)	50 (23.8)	1 (4.2)	10 (21.7)	
Stage									0.441
In situ	65 (16.7)	7 (20.6)	3 (12.0)	7 (24.1)	4 (18.2)	35 (16.7)	1 (4.2)	8 (17.4)	
Localized	158 (40.5)	12 (35.3)	12 (48.0)	11 (37.9)	7 (31.8)	87 (41.4)	17 (70.8)	12 (26.1)	
Regional/distant	66 (16.9)	4 (11.8)	4 (16.0)	5 (17.2)	3 (13.6)	41 (19.5)	2 (8.3)	7 (15.2)	
Surgery									0.467
No	15 (3.8)	3 (8.8)	1 (4.0)	1 (3.4)	0 (0.0)	7 (3.3)	1 (4.2)	2 (4.3)	
Breast-conserving surgery	199 (51.0)	20 (58.8)	18 (72.0)	16 (55.2)	12 (54.5)	102 (48.6)	10 (41.7)	21 (45.7)	
Mastectomy	176 (45.1)	11 (32.4)	6 (24.0)	12 (41.4)	10 (45.5)	101 (48.1)	13 (54.2)	23 (50.0)	
Radiation therapy									0.160
No	209 (53.6)	16 (47.1)	8 (32.0)	15 (51.7)	11 (50.0)	115 (54.8)	13 (54.2)	31 (67.4)	
Yes	170 (43.6)	18 (52.9)	15 (60.0)	14 (48.3)	10 (45.5)	89 (42.4)	11 (45.8)	13 (28.3)	
Chemotherapy									0.008
No	217 (55.6)	21 (61.8)	18 (72.0)	15 (51.7)	9 (40.9)	117 (55.7)	6 (25.0)	31 (67.4)	
Yes	172 (44.1)	12 (35.3)	7 (28.0)	14 (48.3)	13 (59.1)	93 (44.3)	18 (75.0)	15 (32.6)	
ER status									0.347
Negative	54 (13.8)	5 (14.7)	3 (12.0)	4 (13.8)	6 (27.3)	24 (11.4)	6 (25.0)	6 (13.0)	
Positive	319 (81.8)	27 (79.4)	22 (88.0)	22 (75.9)	15 (68.2)	176 (83.8)	18 (75.0)	39 (84.8)	
PR status									0.673
Negative	98 (25.1)	8 (23.5)	6 (24.0)	6 (20.7)	9 (40.9)	51 (24.3)	8 (33.3)	10 (21.7)	
Positive	270 (69.2)	22 (64.7)	18 (72.0)	20 (69.0)	12 (54.5)	148 (70.5)	16 (66.7)	34 (73.9)	
HER2 status									0.082
Negative	253 (64.9)	21 (61.8)	17 (68.0)	20 (69.0)	13 (59.1)	139 (66.2)	14 (58.3)	29 (63.0)	
Positive	60 (15.4)	2 (5.9)	4 (16.0)	1 (3.4)	5 (22.7)	31 (14.8)	9 (37.5)	8 (17.4)	
Fear of recurrence									0.470
Low	132 (33.8)	13 (38.2)	12 (48.0)	8 (27.6)	10 (45.5)	63 (30.0)	10 (41.7)	16 (34.8)	
Moderate	201 (51.5)	19 (55.9)	11 (44.0)	15 (51.7)	11 (50.0)	109 (51.9)	13 (54.2)	23 (50.0)	
High	50 (12.8)	2 (5.9)	2 (8.0)	6 (20.7)	1 (4.5)	33 (15.7)	1 (4.2)	5 (10.9)	

Percentages may add to less than 100% due to rounding and missing data. Other includes participants from countries in Central or South America (Argentina, Brazil, Peru, Honduras, and Panama) and Europe (Spain and Portugal). ER: estrogen receptor; PR: progesterone receptor.

**Table 2 curroncol-32-00461-t002:** Least squares means and 95% confidence intervals of fatalism scores across sociodemographic and clinical characteristics.

Variable	Category	*N*	Estimated Mean (95% CI)	*p*-Value
Total		390	16.4 (15.8–17.0)	
Hispanic origin	Colombian	34	13.3 (11.2–15.4)	0.060
	Cuban	25	15.6 (13.1–18.1)	
	Dominican	29	17.4 (15.1–19.7)	
	Mexican	22	17.8 (15.2–20.5)	
	Puerto Rican	210	16.7 (15.9–17.6)	
	Venezuelan	24	17.4 (14.9–132.8)	
	Other Hispanic	46	15.8 (14.1–17.7)	
Current age (years)	20–<40	15	18.7 (15.5–21.8)	0.377
	40–<55	131	16.6 (15.5–17.7)	
	55–70	159	15.9 (14.9–16.9)	
	≥70	85	16.7 (15.3–18.0)	
Race	White	207	15.9 (15.1–16.8)	0.111
	Non-white	183	17.0 (16.0–17.9)	
Body mass index (kg/m^2^)	<25	109	16.2 (15.1–17.4)	0.709
	25–<30	141	16.3 (15.2–17.3)	
	≥30	135	16.8 (15.7–17.9)	
Current smoking	No	372	16.3 (15.7–17.0)	0.304
	Yes	15	18.1 (14.8–21.3)	
Marital status	Married	234	16.6 (15.7–17.4)	0.770
	Unmarried	151	16.4 (15.3–17.4)	
Household income	<USD 20,000	67	18.5 (17.0–20.0)	0.004
	USD 20,000–<USD 75,000	176	16.1 (15.2–17.0)	
	≥USD 75,000	92	15.1 (13.8–16.3)	
	Prefer not to answer	55	17.1 (15.5–18.8)	
Education level	≤High school	122	18.2 (17.1–19.3)	<0.001
	Some college or more	267	15.6 (14.8–16.4)	
Language use at home	More Spanish	203	17.1 (16.2–17.9)	0.007
	Both equally	86	17.0 (15.6–18.3)	
	More English	99	14.7 (13.5–16.0)	
Birthplace	United States	81	15.8 (14.4–17.2)	0.314
	Outside United States	308	16.6 (15.9–17.3)	
Multimorbidity	No	157	16.7 (15.7–17.7)	0.513
	Yes	218	16.2 (15.4–17.1)	
Cancer stage	In Situ	65	16.2 (14.6–17.7)	0.574
	Localized	158	16.1 (15.1–17.1)	
	Regional/distant	66	17.0 (15.5–18.6)	
Surgery	No	15	13.8 (10.6–17.0)	0.259
	Breast-conserving surgery	199	16.6 (15.7–17.5)	
	Mastectomy	176	16.5 (15.5–17.4)	
Radiotherapy	No	209	16.4 (15.6–17.3)	0.895
	Yes	170	16.4 (15.4–17.3)	
Chemotherapy	No	217	16.3 (15.4–17.1)	0.527
	Yes	172	16.7 (15.7–17.6)	
Estrogen receptor	Negative	54	16.3 (14.6–18.0)	0.881
	Positive	319	16.4 (15.7–17.1)	
Years since diagnosis (years)	<2	42	14.1 (12.2–16.1)	0.043
	2–<5	216	16.8 (16.0–17.7)	
	5–<10	132	16.5 (15.4–17.6)	
Years lived in US	<10	64	16.1 (14.5–17.6)	0.718
	10–<30	119	16.8 (15.6–17.9)	
	≥30	206	16.3 (15.4–17.2)	
Fear of recurrence	Low	132	15.6 (14.5–16.7)	0.039
	Moderate	201	16.6 (15.7–17.4)	
	High	50	18.2 (16.5–20.0)	

**Table 3 curroncol-32-00461-t003:** Correlation matrix among the Multidimensional Fatalism Measures of Hispanic breast cancer survivors (*N* = 390).

Variable	Fatalism	Religious Belief	External Locus of Control	Internal Locus of Control	Belief in Luck
Mean (SD)	16.4 (6.3)	21.3 (7.5)	10.9 (5.3)	23.4 (5.0)	12.6 (4.7)
Correlation Matrix: Correlation Coefficient (*p*-value)					
Religious belief	0.394 (*p* < 0.001)				
External locus of control	0.400 (*p* < 0.001)	0.125 (*p* = 0.014)			
Internal locus of control	0.150 (*p* = 0.003)	−0.003 (*p* = 0.949)	−0.045 (*p* = 0.677)		
Belief in luck	0.273 (*p* < 0.001)	0.156 (*p* = 0.002)	0.384 (*p* < 0.001)	0.060 (*p* = 0.234)	

**Table 4 curroncol-32-00461-t004:** Results from a multivariable linear regression model, investigating factors associated with fatalism score among Hispanic breast cancer survivors (*N* = 390).

Characteristics	Parameter	b	SE	*p*-Value
	Intercept	19.2	2.4	<0.0001
Hispanic origin (Ref: Puerto Rican)	Colombian	−3.7	1.1	0.001
	Cuban	−0.9	1.3	0.509
	Dominican	−0.1	1.2	0.920
	Mexican	0.7	1.4	0.622
	Venezuelan	1.4	1.4	0.314
	Other Hispanic	−0.7	1.0	0.480
Current age (Ref: 20–<40)	40–<55	−1.9	1.7	0.264
	55–<70	−3.0	1.7	0.082
	70+	−3.3	1.8	0.069
Race (Ref: white)	Non-white	0.7	0.6	0.289
Household income (Ref: <USD 20,000)	USD 20,000–<USD 75,000	−2.4	0.9	0.007
	≥USD 75,000	−2.4	1.1	0.029
	Prefer not to answer	−1.2	1.1	0.273
Education (Ref: ≤ high school)	Some college+	−2.0	0.7	0.008
Language use at home (Ref: more Spanish)	Both equally	−0.2	0.9	0.804
	More English	−1.9	0.9	0.043
Years lived in US (Ref: <10)	10–<30	0.9	1.0	0.380
	30+	1.6	1.0	0.110
Years since diagnosis (Ref: <2)	2–<5	2.0	1.0	0.061
	≥5	2.1	1.1	0.052
Fear of recurrence (Ref: low)	Moderate	0.8	0.7	0.225
	High	1.8	1.1	0.103

b: regression coefficient, SE: standard error.

## Data Availability

Restrictions apply to the availability of the cancer registry’s data. Data were obtained from the Florida Cancer Data System and are available with the permission of the Florida Department of Health and Florida Cancer Registry. The survey data presented in this article are not readily available because the data are part of an ongoing study. Requests to access the survey data should be directed to the corresponding author.

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
