# Peer review of "Evaluating Fatalism Among Breast Cancer Survivors in a Heterogeneous Hispanic Population: A Cross-Sectional Study"

_curroncol, 2025, doi:10.3390/curroncol32080461_

Round 1

Reviewer 1 Report

Comments and Suggestions for Authors

The manuscript was clearly written and easy to follow. The theme of the study is interesting and will contribute to literature with new insights.

Minor comments:

  1. Are the Fatalism scores normally distributed? If not, in the multivariate linear model, are the residual terms normally distributed with zero means? A graphic examination suffices.
  2. Just for an information, the Kruskal-Wallis test is a tool for normality data when comparing means across various groups.
  3. In the Result section, the authors reported that “Most of them were 200 diagnosed at an early stage with ER-positive tumors.” Breast cancer has three markers: Estrogen receptor (ER), Progesterone receptor (PR) and human epithelial growth factor-receptor 2 (HER2). Information on PR and HER2 would be beneficial for readers.
  4. For missing values, the listwise-deletion was adopted, namely, “only the cases with non-missing data were included in the multivariable analysis.” Did the authors consider a multiple imputation approach could serve as a sensitive analysis?

Reviewer 2 Report

Comments and Suggestions for Authors

The authors present the results of an analysis done to evaluate the mediators of the QOL disparities among breast cancer survivors within the Hispanic community in Central Florida.

The article is well-written, and the language is concise.

Specific comments by section and line number follow.

Introduction

51: Remove "the"

59: remove extra space

Materials and Methods

82: remove "the"

155: Do you mean locale instead of study?

158: remove "their"

164: Were household income and education level individual measures or small-area measures? If all characteristics were individual level, please indicate.

Results

212: remove "were"

Table 2: Please tighten this table to enhance readability. One option would be to list categories slightly indented under variables. For instance:

Hispanic origin

     Columbian

      Cuban

      etc.

This would give you more space width-wise and also force a gap between each variable type.

Discussion

289: replace ", which could be because that" with "due to"

290: remove "them"

364: While you indicated earlier in the discussion that "Future studies should...", please indicate what your next steps are in regard to this research/findings.
